# Herpesvirus Infection in a Breeding Population of Two Coexisting *Strix* Owls

**DOI:** 10.3390/ani11092519

**Published:** 2021-08-27

**Authors:** Zoran Žlabravec, Al Vrezec, Brigita Slavec, Urška Kuhar, Olga Zorman Rojs, Joško Račnik

**Affiliations:** 1Institute of Poultry, Birds, Small Mammals, and Reptiles, Faculty of Veterinary Medicine, University of Ljubljana, Gerbičeva ulica 60, 1000 Ljubljana, Slovenia; zoran.zlabravec@vf.uni-lj.si (Z.Ž.); brigita.slavec@vf.uni-lj.si (B.S.); olga.zormanrojs@vf.uni-lj.si (O.Z.R.); 2National Institute of Biology, Department for Organisms and Ecosystems Research, Večna pot 111, 1000 Ljubljana, Slovenia; al.vrezec@nib.si; 3Slovenian Museum of Natural History, Prešernova cesta 20, 1000 Ljubljana, Slovenia; 4Institute of Microbiology and Parasitology, Faculty of Veterinary Medicine, University of Ljubljana, Gerbičeva ulica 60, 1000 Ljubljana, Slovenia; urska.kuhar@vf.uni-lj.si

**Keywords:** wildlife, predator-prey interaction, disease transmission, Ural owl, tawny owl, yellow-necked mouse, polymerase chain reaction, Slovenia

## Abstract

**Simple Summary:**

Although infection with herpesvirus in owls has commonly been described as a highly lethal disease, there is very little information about the presence of herpesvirus and its potential impact in living owls in wild populations. Our study detected herpesvirus in a breeding population of Ural owls, which showed no clinical signs of illness nor productivity deviances (i.e., in clutch and brood size). Herpesvirus was detected in Ural owl adults and chicks, but not in a young tawny owl (despite the fact that they were in same nest and in persistent contact). Furthermore, herpesviruses were also detected in yellow-necked mice as both owls’ main prey. However, comparison of the herpesviruses detected showed that different herpesviruses are present in the owls and mice. The results of this study show that herpesvirus may be present in a Ural owl breeding population without any consequences on health and breeding performance. However, in the case of tawny owls, it seems that they are not susceptible to infection, which could be related to their polymorphism. It seems that small rodents are not a source of herpesvirus infection in owls and that the probable herpesvirus transmission pathway takes place intraspecifically, mostly from adults to young.

**Abstract:**

Birds are a frequent host of a large variety of herpesviruses, and infections in them may go unnoticed or may result in fatal disease. In wild breeding populations of owls, there is very limited information about the presence, impact, and potential transmission of herpesvirus. The herpesvirus partial DNA polymerase gene was detected using polymerase chain reaction in oropharyngeal swabs of 16 out of 170 owls examined that were captured in or near nest boxes. Herpesvirus was detected in Ural owls (*Strix uralensis*), in both adults and young, but not in tawny owls (*Strix aluco*). In yellow-necked mice (*Apodemus flavicollis*), as the main prey of tawny owls and Ural owls in the area, herpesvirus was detected in the organs of 2 out of 40 mice captured at the same locations as the owls. Phylogenetic analysis showed that the herpesvirus sequences detected in the Ural owls differed from the herpesvirus sequences detected in the yellow-necked mice. The results indicate that herpesvirus infection exists in the breeding wild Ural owl population. However, herpesvirus-infected owls did not show any clinical or productivity deviances and, based on a phylogenetic comparison of detected herpesvirus sequences and sequences obtained from Genbank database, it seems that mice and other rodents are not the source of owl infections. The most probable transmission pathway is intraspecific, especially from adults to their chicks, but the origin of herpesvirus in owls remains to be investigated.

## 1. Introduction

Some pathogens pose significant natural hazards for wild bird populations [1], and they even have potentials for outbreaks in humans, especially when ecosystems and thus regulatory ecosystem services are depleted [2]. Raptors, as predators at the top of the food chain, are particularly good environmental sentinels for detection of wildlife zoonosis [3]. However, the prevalence, transmission, and impacts of viruses in free-ranging raptors are still a poorly understood phenomenon, and this is probably reflected in cumulative effects in raptor mortality and fecundity combined with other environmental impacts [4,5], as well as individual variation [6].

Diverse herpesviruses have frequently been found in different free-living bird species [7,8,9,10,11]. In owls, herpesvirus was discovered in the 1970s [12], later known as Strigid herpesvirus (StHV 1), and it has been reported in captive and free-ranging owls in Asia, Europe, North America, and Australia [10]. General, all avian herpesviruses are members of the genera *Iltovirus* and *Mardivirus* of the subfamily Alphaherpesvirinae. However, many viruses detected in wild birds have not been completely characterized, including StHV 1, and therefore has not been approved as species and are marked as other related viruses which may be member of the family Herpesviridae [13,14].

The clinical signs of herpesviruses infections in owls, known as inclusion body disease or herpesvirus hepatitis, appear as general depression and anorexia before death or sudden death [15]. In addition, more specific clinical signs such as ulcerative superficial keratitis, proliferative conjunctivitis, and iris pigmentary changes have been described [16]. Hepatitis and disseminated focal necrosis in the liver, spleen, and bone marrow are most commonly seen in owls dying of herpesvirus infections [13]. The actual impact of herpesvirus on the wild owl population became apparent in cases of repopulation of eagle owls (*Bubo bubo*) where serologically herpesvirus-negative birds released back to the wild led to the establishment and expansion of the population [17]. Earlier studies of owl herpesvirus in Europe reported that, on the one hand, the virus can be lethal to some owl species but non-infective to some others, especially species with dark eyes (i.e., the tawny owl (*Strix aluco*) and barn owl (*Tyto alba*) [18,19]), while the third European owl species with dark eyes, the Ural owl (*Strix uralensis*), was not tested. A later study, which included Ural owls as well, did not confirm this hypothesis but indicated that polymorphic species are seemingly more resistant than the species showing lower variability in overall plumage color [11]. However, all of these studies were conducted on dead or injured birds from wildlife rehabilitation centers or birds in laboratory experiments rather than in free-living wild populations.

The viral DNA polymerase is a key enzyme in the lytic phase of the infection by herpesviruses [20]. PCR assays [21,22] with degenerate primers for amplification of the herpesvirus DNA polymerase gene sequence have shown tremendous effectiveness in detecting previously unknown herpesvirus [23]. Based on a partial herpesvirus DNA polymerase sequence phylogenetic study showed that herpesvirus in owls and herpesvirus endemic in the pigeon population—namely, *Columbid herpesvirus 1* (CoHV 1)—are the same virus, and that the pigeons are responsible for transmission of the virus to the owls [7,24]. However, a recent study of phylogenetic analysis of herpesvirus DNA polymerase partial nucleotide sequences detected in dead owls showed that owls were also infected with herpesviruses that are divergent from CoHV 1 [11]. In general, inhalation of virus-containing dust derived from feathers, nasal excretion, saliva, nasal discharge, urine, feces, and crop milk is the predominant means of herpesvirus transmission in birds, and no vertical transmission has been proven [25].

To the best of our knowledge, this study is the first on herpesvirus in owls conducted in the wild breeding owl population. We studied two closely related, ecological similar, and coexisting owl species, the Ural owl and tawny owl [26], which were found to have different susceptibility to herpesvirus infection in previous study on dead birds [11]. In Europe, the Ural and tawny owl largely overlap in range, habitat, prey, and nest site selection [26,27,28], indicating close contact between the species and thus high exposure to interspecific virus transmission. The larger Ural owl can also outcompete the smaller tawny owl through direct interspecific territoriality and even predation [27,29]. Thus, our aim was to ascertain whether herpesvirus is present in free-living owls, to determine its potential impact on clinical health and breeding parameters taking into account adult breeding birds and young of both species, and to characterize herpesviruses in line with previously detected herpesviruses. We predicted two possible means of virus transmission, namely via prey (the main prey of both species are mice and voles [30,31]) or via intra- and interspecific direct contact between interacting owls.

## 2. Materials and Methods

### 2.1. Field Sampling of Strix Owls

Field sampling was conducted between 2017 and 2019 on Mount Krim (45°58′ N, 14°25′ E; central Slovenia) and the Jelovica Plateau (46°18′ N, 14°8′ E; northern Slovenia), which are large mixed forest areas dominated by beech (*Fagus*), fir (*Abies*), and spruce (*Picea*) at elevations between 300 and 1600 m [32]. Tawny and Ural owls coexist in relatively high breeding densities [33] in the area, and they were found to compete for space, nest sites, and food [26]. In both areas, a network of nestboxes for large owls was established to allow detailed study of both species’ biology. We trapped adult breeding birds and their young in or near the nestboxes during annual nest inspections and ringing under license no. 3561-40/2017-4, issued by the Slovenian Environment Agency. Each bird was clinically examined, and oropharyngeal and cloacal swabs were collected from 43 breeding adults and 127 young of the tawny owl and Ural owl. No birds were harmed during the sampling procedure, and all 170 owls were released at the site of capture soon thereafter. Dry FloqSwabs (Copan Italia SpA, Brescia, Italy) were immediately placed in sealed plastic at 4 °C, transported to the laboratory, and stored at −20 °C until analyzed. In each sampled nest we also recorded clutch (no. eggs per nest) and brood size (no. of young per nest) as a measure of productivity.

### 2.2. Field Sampling of Mice

The main prey of tawny and Ural owls in the area are forest-dwelling rodents, especially the yellow-necked mouse (*Apodemus flavicollis*) [32,34], which is one of the dominant small mammal species in temperate forests [35]. Mice were sampled in 2019 with snap traps baited with a mixture of canned sardines and rolled oats in both areas, two sampling locations at low and high elevations, in the forest at the end of the owl breeding season in June, comprising on average 122 ± 43 trap-nights each year.

Forty mouse carcasses were kept individually in plastic bags and stored at −20 °C at the Institute for Poultry, Birds, Small Mammals, and Reptiles at the University of Ljubljana’s Veterinary Faculty, where partial necropsy was performed. During the necropsy, oropharyngeal and rectal swabs, and lung, kidney, liver, spleen, and brain samples were taken from each carcass using cleaned and autoclaved instruments. A different set of instruments was used for each necropsy to prevent DNA contamination. Tissues were either processed immediately or stored at −70 °C until use.

### 2.3. DNA Extraction and PCR of a DNA Polymerase Gene Region Using Herpesvirus Consensus Primers

Oropharyngeal and cloacal swabs collected from free-living owls and oropharyngeal and rectal swabs from free-living rodents were individually vortexed in 2 mL phosphate-buffered saline for 2 min. Oropharyngeal and cloacal/rectal swabs were separately pooled per five samples together; 100 µL aliquots of each swab in PBS were pooled to produce 500 µL samples for genomic nucleic acid extraction. Tissue samples from individual free-living rodents were pooled and homogenized in phosphate-buffered saline in ratio 1:10 volume/volume. The homogenates were clarified by centrifugation at 1000× *g* for 10 min.

Total DNA and RNA were extracted from 140 µL from pooled samples by the QIAamp Viral RNA Mini Kit (Qiagen, Hilden, Germany) according to the manufacturer’s instructions. Viral DNA was detected by nested PCR using a set of degenerate primers that target the herpesvirus DNA polymerase gene region as described by VanDevanter et al. [21]. The PCR volume was 20 µL, and it contained 10 µL of DreamTag Green PCR Master Mix (2×; Thermo Scientific, Dreieich, Germany), 1 µM of each PCR primer, 2 µL of isolated DNA, and deionized water up to 20 µL. Cycling parameters for both methods of PCR used an initial denaturation at 95 °C for 5 min, followed by 45 cycles of 94 °C for 30 s, 46 °C for 60 s, 72 °C for 60 s, and final extension at 70 °C for 7 min. Individual samples from positive pools of samples were tested individually, as previously described.

### 2.4. Detection, Sequencing, and Phylogenetic Analysis of PCR Products

PCR products were analyzed in 1.8% agarose gel (Sigma-Aldrich, St. Louis, MO, USA) containing ethidium bromide by electrophoresis, purified using a FastGene Gel/PCR extraction kit (Nippon Genetics, Duren, Germany) and sent for sequencing to the Macrogen Laboratory (Macrogen Inc., Amsterdam, The Netherlands). The nucleotide sequences obtained were first analyzed by BLAST [36] to identify sequences relevant for further analyses within the NCBI database. Nucleotide sequences were aligned in Geneious with MAFFT translation alignments [37]. Phylogenetic analysis was performed using the maximum likelihood method with the Tamura 3-parameter plus gamma distribution plus invariable site and 1000 bootstrap replicates by MEGA 7.0 [38]. The percentage of similarity among sequences was calculated by the p-distance model (pairwise distance) in MEGA 7.0. The accession numbers of the herpesvirus sequences obtained in this study are MW315868–MW315883 and MW345631–MW345632. The accession numbers of other herpesvirus sequences used for phylogenetic analysis are included in Figure 1.

## 3. Results

### 3.1. Herpesvirus Detection in Owls and Yellow-Necked Mouse Populations

The partial sequence of herpesvirus DNA polymerase gene was detected only in oropharyngeal swabs in 16 out of 170 owls examined (9.4%). However, the herpesvirus was detected only in Ural owls, in adults and young, but not in tawny owls (Table 1). Out of 45 Ural owls from Mount Krim, 14 were positive (31.1%), and out of 10 Ural owls from the Jelovica Plateau, 2 were positive (20.0%), but the infection prevalence was not significantly different between the areas (χ² = 0.11, *p* = 0.74). The infection prevalence was found to be higher in young than adult birds, although the difference was not significant (χ² = 0.59, *p* = 0.44). We found that about half of the Ural owl nests were infected (Table 1), but we did not necessarily confirm infection in all birds from each infected nest. The herpesvirus prevalence was found to be similar in adult females (22.2%, *n* = 9) and males (16.7%, *n* = 6). In the nests with more than one young, of which at least one was positive (*n* = 5 nests), the median herpesvirus prevalence in the young was 50.0% (range 20.0 to 100.0%). In the nests with tested breeding female and chicks, all chicks were infected when female was positive (*n* = 2 nests), while positive chicks were found also in the nests with females found negative for herpesvirus infection (*n* = 4 nests). In one nest in 2019 on Mount Krim, the Ural owl pair successfully raised one Ural and one tawny owl young, which was a consequence of competitive expulsion of the tawny owl by the larger Ural owl from the nestbox. Both Ural owl parents were herpesvirus positive as well as the Ural owl young, but not the tawny owl young raised in the same nest. All the owls sampled were clinically healthy, and the productivity and clutch and brood size between infected (median 4.0/3.5) and uninfected nests (median 3.5/3.0) was similar (Mann–Whitney *U* = 20–23, *p* = 0.6–1).

In 2019, 40 yellow-necked mice captured at Mount Krim and the Jelovica Plateau were examined for the presence of herpesvirus. Pharyngeal and rectal swabs and tissue samples were analyzed with panherpesvirus PCR targeting the DNA polymerase gene. Of 40 free-living mice examined, 2 were herpesvirus positive (5%).

### 3.2. Phylogenetic and Sequence Analysis

The partial nucleotide herpesvirus sequences (205 nt) detected in wild Ural owls and yellow-necked mice were compared to the sequences of DNA polymerase gene of other avian and mammal herpesviruses to determine their phylogenetic relationship.

The phylogenetic analyses showed that the herpesvirus sequences detected in Ural owls are most closely related to the alphaherpesvirus sequences detected in other owls, whereas the herpesvirus sequences detected in small rodents formed a separate group among betaherpesviruses (Figure 1). More precisely, the sequences of herpesviruses detected in owls clustered together with novel herpesvirus sequences detected in the Eurasian eagle owl (Bubo bubo HV), Ural owl (Strix uralensis HV), long-eared owl (Asio otus HV), and great horned owl (StHV 1). The phylogenetic tree (Figure 1) showed that partial DNA polymerase gene sequences of wild Ural owls from Slovenia were identical and showed 100% nt identities with the herpesvirus sequence detected in the long-eared owl and Ural owl, and 90.6% and 90.0% nt identities with the Eurasian eagle owl and great horned owl, respectively. Known herpesvirus sequences detected in pigeons (CoHV 1) and birds of prey (CoHV 1, FaHV 1, GaHV 2) were grouped in the other clusters, and they shared 60.3% to 62.2% nt identities with novel herpesvirus sequences detected in Ural owls.

The DNA polymerase sequences detected in yellow-necked mice were identical and shared 98.7% nt identity with the most closely related betaherpesvirus sequence detected in yellow-necked mice (Apodemus flavicollis cytomegalovirus 3) from Germany. Lower nt identities, 47.2 to 48.4%, were detected with other betaherpesvirus sequences in the yellow-necked mouse (Apodemus flavicollis cytomegalovirus 2 and 1), fawn-colored mouse (Mus cervicolor cytomegalovirus 1), and wood mouse (Apodemus sylvaticus cytomegalovirus 1).

There was low nt identity (51.1%) between the herpesvirus sequence detected in wild mice and owls in Slovenia.

## 4. Discussion

Based on the herpesvirus DNA polymerase gene detected in wild owls, the results of our study showed that herpesvirus infection exists in breeding wild owl populations, but only the Ural owl appeared to be a herpesvirus reservoir. However, the fact that no herpesviruses were detected in tested tawny owls should be taken with caution, as several factors may influence the detection of herpesvirus, such as the small number of birds tested, viral shedding at the time of sampling, pooling of samples, and the sensitivity and specificity of the PCR method. All of these factors can influence on negative results, especially when virus levels in the sample are low or in the case of novel viruses. Infected birds did not show any clinical or productivity deviances, despite the fact that infection with herpesvirus is known as a fatal disease in owls [15,25], but the effects might be evident in long-term productivity and survival, especially in low-prey seasons [4,39]. Based on the herpesvirus DNA polymerase gene detected in oropharyngeal swabs in clinical healthy wild owls, this could imply that asymptomatic shedding of the virus was detected. Herpesviruses have the ability to establish lifelong latency within the host and to periodically reactivate [25], and it is not unusual that an individual might be carrying the pathogen without any visible manifestation of the disease until exposure to an environmental stressor triggers activation of viral replication and spread to a new host organism. Shedding of herpesvirus from the oropharyngeal cavity of birds is an important factor from the point of view of potential virus transmission from infected to uninfected individuals. In owls, the male commonly offers food to the female during the courtship ritual and, furthermore, the male and female both feed their offspring during the reproduction period [27]. The results showed that oropharyngeal swabs are more suitable for detection of herpesvirus in owls; however, although herpesvirus was not detected in cloacal swabs, which could also be due to the presence of inhibitors in the swabs, the importance of cloacal swabs in the detection of herpesvirus should not be neglected, as previous reports have shown [23,40]. Our study was conducted during peak prey years [41], and thus with low nutritional stress for owls, but even here transmissions within the nests were detected. We speculate that during low-prey seasons herpesvirus infections might have more detrimental effects on Ural owls. Such infections are known to completely suppress breeding in low-prey seasons, which is not the case in tawny owls [41].

In general, herpesviruses are widespread, with various clinical manifestations in several avian species throughout the world [42]. However, neither transmission nor pathogenesis are fully understood in free-living birds. Although that some studies have suggested that raptors may contract the infection and consequently disease through the oral route by ingesting CoHV 1 infected pigeons [7], the detection of the partial sequence of herpesvirus DNA polymerase gene in healthy owls raises the question whether all herpesvirus infections in this species [7,24] cause illness with a fatal outcome in some cases. GaHV 2, also known as Marek’s disease virus [11], CoHV 1 [7,10,11], StHV 1 [16], and the herpesviruses detected in our previous [11] and present study are currently known herpesviruses in owls. They are genetically different, and it could be possible that some of them, such as CoHV 1, are a more fatal threat to owls than others, and also that they could differ from each other in transmission and pathogenesis. For more complete information on the potential pathogenicity of herpesvirus in owls, a virus isolation, serotyping, and full genome characterization of these different herpesviruses should be performed in the future.

The herpesvirus sequences detected in Ural owls were grouped together with herpesvirus sequences detected in a dead free-living Eurasian eagle owl, Ural owl, and long eared owl (*Asio otus*) from our previous study [11], and in live free-living great horned owl (*Bubo virginianus*) from the United States [16], suggesting that this herpesvirus is spread transcontinental. Although, it seems that this herpesvirus infects only species within group Striginae, which is a monophyletic clade [43] with owls from the genus *Bubo* and *Strix* being most closely related [44], what additionally supports our and previous findings. However, our study revealed that even within Striginae herpesvirus susceptibility is not equal. Within the genus *Strix,* tawny and Ural owls were found to be closely related species [43], but our study showed that species differ significantly in herpesvirus susceptibility, however a possibility of other herpesviruses presence should be emphasized, which could be missed out by used PCR method. We believe that a simple correlation between herpesvirus and the owl molecular phylogeny is insufficient to explain the host-parasite relationship. Possible reasons in apparently greater immunity of the tawny owl against herpesvirus infection may lay in owl metabolic traits, as suggested in a previous study [11] or in recent co-evolutionary processes that differed between different owl hosts and herpesvirus. Both owl species are highly diverged in life traits. In our study area, the Ural owl is regarded as a glacial relict species with a major part of its population found in boreal climate zone of Eurasia, while the tawny owl is a temperate species that has recently spread to the Northern Europe [45,46,47]. Our study was confined to a small geographic scale within temperate climate zone, where both species coexist and where Ural owl reach its southern limit of distribution [32]. Future studies should take into account larger geographical scale of herpesvirus sampling in tawny and Ural owl wild populations across Europe to study climate and trait related impacts of herpesvirus infection prevalence in both *Strix* species, suggesting that populations at the limit of their distribution may be more susceptible to infections.

Although host-virus co-evolution is thought to be the primary mode of herpesvirus evolution, cross-species transmission events have been known to occur [7,48,49]. Phylogenetic analysis of the partial sequence of herpesvirus DNA polymerase gene showed identical sequences within both positive animal species, in Ural owls and yellow-necked mice. The herpesviruses detected in mice were different from the herpesvirus sequences detected in owls. The complete natural host range of most herpesviruses is not known, and it appears that the range of natural hosts is very narrow for some herpesviruses, but others affect many different bird species (Figure 1). A herpesvirus that infects both mammals and birds has been rarely described. To our knowledge, Suid herpesvirus 1, causative agent of pseudorabies also named as Aujeszky’s disease which belongs to alfaherpesviruses is the only herpesvirus that causes generally mild disease in swine and severe disease including mortality in some other mammals, and also in chickens or pigeons; however, herpesvirus transmission from mammals to birds was reported only under experimental conditions [50]. The herpesvirus DNA polymerase sequences detected in mice in this study were clustered with cytomegalovirus 3 (CMV 3) (subfamily betaherpesviruses) detected in yellow-necked mice in Germany. Other herpesviruses were also detected in the same host; namely, CMV 1, CMV 2, *Apodemus flavicollis rhadinovirus 1* (AflaRHV1), and *Murine gammaherpesvirus 68*, which were clustered in other groups among betaherpesviruses and gammaherpesviruses [51]. In this large-scale study, with over 1100 samples of blood and tissue samples, many known and novel beta- and gamma-herpesviruses were also detected that are found in other species of free-living rodents, such as the bank vole (*Clethrionomys glareolus*), field vole (*Microtus agrestis*), common vole (*Microtus arvalis*), long-tailed field mouse (*Apodemus sylvaticus*), house mouse (*Mus musculus*)*,* Norway rat (*Rattus norvegicus*)*,* and black rat (*Rattus rattus*). Similar to our study, no evidence of alphaherpesvirus was obtained. Low prevalence, unknown loci of latency, low viral loads of samples, or the hypothesis that alphaherpesvirus in rodents never developed or became extinct during herpesvirus evolution were some of the reasons that were proposed [51].

On the other hand, no detected rodent herpesviruses were found in owls, despite the fact that they represent a large part of their diet. The herpesviruses detected in owls are currently unassigned in the family Herpesviridae. However, phylogenetic analysis of DNA polymerase sequences detected in owls in Slovenia and North America shows that they are most closely related to subfamily alphaherpesvirus. Betaherpesviruses and gammaherpesviruses are known for their biological characteristic of restricted host range [52], which could be one of the reasons why no rodent herpesviruses (betaherpesviruses and gammaherpesviruses) were detected in owls. Nevertheless, because of the large proportion of rodent diet in owls and relatively sensitive PCR method, the detection of rodent herpesvirus in the oropharyngeal region and cloaca would be expected, at least as part of contamination with herpesvirus. To conclude, based on our results and the results presented in the herpesvirus study by Ehlers et al. [51], it seems we can rule out the prey–predator herpesvirus transmission pathway since there was very low similarity between the herpesvirus sequences detected in owls and mice and also other rodents. However, due to the high level of diversity among herpesviruses it should be emphasized that used consensus PCR method could miss a novel herpesvirus occurring [23]. The strongest and probably the most important transmission pathway is intraspecific, especially from adults to their chicks. However, within the nest, transmission among the chicks seemed to be lower.

We confirmed the conclusions of previous studies on dead and experimental birds [11,18] that tawny owls are not susceptible to herpesvirus, not even when in close contact with infected Ural owls. However, there is still a pending question about lack of susceptibility of tawny owls to herpesvirus, although tawny and Ural owls are closely related species [43]. Žlabravec et al. [11] suggested polymorphism and greater genetic and physiological variability as a possible reason for tawny owl resistance to infection, but there might be other historical reasons, with the tawny owl being an ancient and coevolved herpesvirus host, whereas infections in Ural owls might be relatively recent, which should be explored further in future studies.

## 5. Conclusions

In conclusion, this study has provided greater insight into the herpesvirus presence in free-living owls, their epizootiology, and the wildlife impact on the breeding population of owls in temperate forests in Europe. Herpesvirus was detected in live free-living breeding population of Ural owls, but not in tawny owls. The herpesviruses detected in owls were phylogenetically identical and were most similar to herpesvirus sequences detected in organs of dead free-living owls from our previous study. However, comparison with the herpesvirus detected in yellow-necked mice and other rodents ruled out small rodents as a possible source of herpesvirus infection in owls. It seems that most important herpesvirus transmission occurs from adults to young, but the origin of the herpesvirus detected in owls remains to be further investigated. Nevertheless, despite the fact that herpesvirus in owls was commonly described as a fatal disease, no clinical signs or productivity deviances were recorded in breeding population of herpesvirus-positive owls.

## Figures and Tables

**Figure 1 animals-11-02519-f001:**
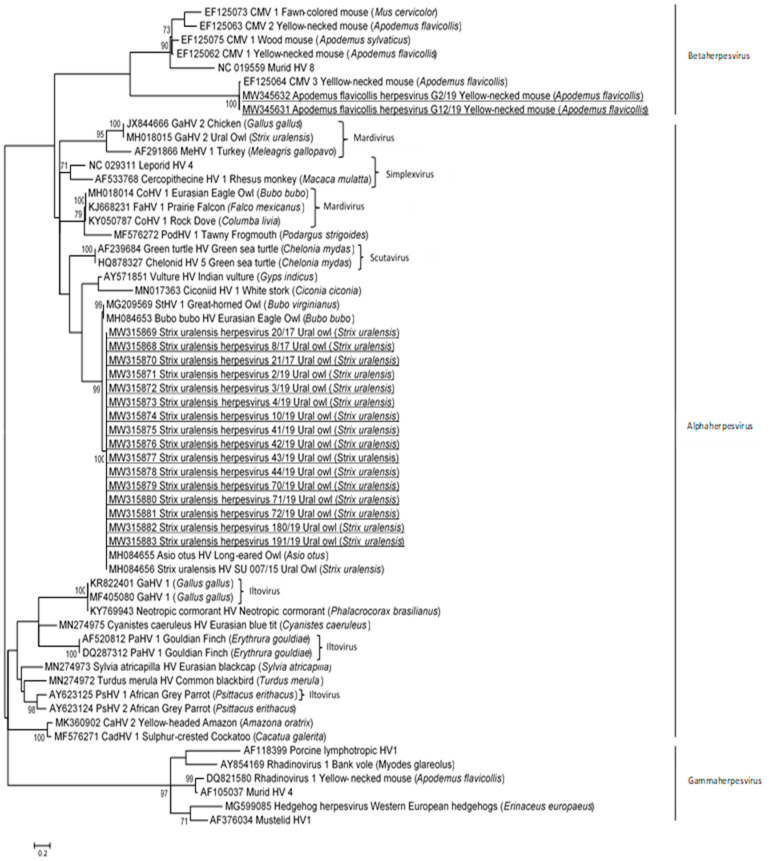
Phylogenetic relationship based on partial DNA polymerase gene nucleotide sequences of herpesviruses from Ural owls and yellow-necked mice captured in Slovenia and other herpesviruses derived from the GenBank database. The scale bar indicates substitutions per site. Nucleotide sequences obtained in the current study are additionally underlined.

**Table 1 animals-11-02519-t001:** Herpesvirus prevalence (% of infected among all tested birds/nests) in wild coexisting populations of tawny owls *(Strix aluco)* and Ural owls *(Strix uralensis)* in Slovenia in 2017 and 2019. The numbers of birds or nests examined are given in parentheses.

	Tawny Owl	Ural Owl
Infected birds	0.0% (115)	29.1% (55)
Infected adults	0.0% (27)	18.7% (16)
Infected young	0.0% (88)	33.3% (39)
Infected nests	0.0% (30)	53.0% (17)

## Data Availability

Sequence data obtained in this study were deposited in the NCBI database under accession numbers MW315868–MW315883 and MW345631–MW345632.

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
