# Peer review of "Herpesvirus Infection in a Breeding Population of Two Coexisting Strix Owls"

_animals, 2021, doi:10.3390/ani11092519_

Round 1
Reviewer 1 Report
Zlabravec: Herpesvirus infection in a breeding population of two coexisting Strix Owls
The authors aimed to understand the viral burden posed by herpesviruses in two different owl species. Overall and interesting study. The introduction suffers from some major virus taxonomy errors and should be corrected to reflect the current Herpesvirus taxonomy. There are some clarity issues as well, but these should be easily resolved. Overall, an interesting paper.
Line 45. This is a bit confusing as it is currently written. I don’t think bird “diseases” have risk for humans, but the causative agents of those diseases do. So perhaps change “diseases” -> “pathogens”, and use the word “zoonotic” in the second sentence.
Line 53. What do the authors mean by “in a large variety”? Do they many a number of different viral species have been detected?
Line 53. The Latest update of the ICTV does not recognise Herpesvirus strigis as a viral species, so this will have to be amended. The reference here [12] is a paper from 1973. Based on the ICTV page and NCBI taxonomy, all the viruses from owls are “unclassified Alphahesperviruses”
Line 56. Based on the above paragraph, I think that rather that discussing the herpesviruses that infect owls, so HV is probably not the correct abbreviation here.
Line 61/62/63. This statement is unclear. Please rephrase. Do the authors imply that if sero-positive owls were released that the population of eagle owls would not expand?
Line 65. Is “dark-eye owl species” an official classification. Ie, a genera? Or do the authors mean nocturnal owls?
Line 67. What do the authors mean by “polymorphic species”. Which trait are they referring to?
Line 74-76. Does the ICTV classify this as the same viral species or not? In fact, now that I am looking at the latest update of the ICTV, there is no viral species called “Strigix herpesvirus”, so the introduction needs to be amended to correct this. Furthermore, based on a blast comparison of any of the owl hesperviruses and coloumbid herpesviruses, the percentage similarity falls outside the range for species inclusion
https://talk.ictvonline.org/ictv-reports/ictv_9th_report/dsdna-viruses-2011/w/dsdna_viruses/91/herpesviridae
Line 94. Given herpesviruses have very much co-diverged with their hosts, I wonder why the authors hypothesized owls are frequently infected by their prey – ie, the assumption of regular cross species transmission against co-phylogeny evolutionary patterns? Further, the existing owl sequnces fall into a distinctively avian clade and not a mammalian clade. More justification is therefore needed as to why this hypothesis was made given it is not parsimonious with the data at hand.
Why were oropharangeal swabs collected. Is this a virus of the lungs? Do we know where these viruses preferentially infect and replicate? What about cloacal or faecal swabs?
Line 113. While DNA viruses are more stable than RNA viruses, I wonder what the effect of sample storage had on detection efficiency. Also, were the swabs placed in some type of media? Eg DNASheils, PBS, UTM?
Line 141. Does this PCR assay use degenerative primers?
Line 155. Clustal W is known to be a poor alignment tool. Can the authors confirm that there were not gaps in the alignment? If there were any gapped sites, a different aligment tool should be used.
Line 157. Why did the authors used Kimura 2 model? The best sub model should be used, and this is selected following a model test. Did the authors perform a model test?
Line 156. Given the phylogenetic tree presented is of the whole family, I wonder if a tree of amino acids would be more appropriate? Or perhaps a good compliment.
Line 175-177. This sentence is a bit confusing. I understand the bit wherein if 1 adult is infected, all the chicks are infected. But I don’t really understand the latter part. Do you mean nests either had 0% infected, or 50-100% infected. What was the infection status of the adults in these nests with at least 50% infection?
Table 1: Please add confidence intervals. Im not entirely convinced this table is the best way to show the data. Maybe a bar/point chart with confidence intervals (and with the same sizes written above each bar).
Figure 1. I suggest the following aesthetic changes: Please make the text of the tips much bigger – it is impossible to read. Even more impossible is the support values on the nodes. Can the auhors also add relevant viral genus and species names. Why was the tree rooted against the Gammherpesviruses?
Line 298. I would ague this study did not provide evidence of “latent infection”, only PCR positive.
Line 298/299. I do not believe that your study excluded infection in Tawny Owls, only in the subset of owls you tested. Specifically, the assay may not have been adequate at detecting infection, sample storage may have been an issues, or Tawny owls have higher prevalence at different locations or times of the year. Just because you found 0/115 does not mean that Tawny owls are not infected with this virus.
Line 300. How was “productivity” measured in this study? There is no mention of this in the results.
Line 300. HV is known to be fatal in some owls in some situations. Importantly, as stated in the intro, this virus has only been characterized in owl in rescue centre previously, which means that there is a good change something else killed the owl, which just happened to also be positive for HV.
Line 308. Why do the authors assume oral shedding for this virus? I don’t believe any transmission route and replicative site work have been done??
Line 328. There have been a number of articles recently published documenting that wild birds may be infected with a multitude of viruses with no disease signs. Further, viruses causing disease may be the outliers (ie. poultry are new host niches in evolutionary time and many viruses that cause disease in poultry don’t cause disease in wild birds, which may be the natural reservoir). Thus, it is likely that wild birds co-evolved with their own herpesviruses and thus we should not expect disease in owls infected with owl herpesviruses (unless the birds are highly stressed or suffering from a coinfection).
Line 331-334. This section could be expanded as it is arguably the most interesting.
Line 337-338. Does this sentence mean you found identical sequences in mice and owls? Based on the tree, I would argue this is certaintly not true. Therefore, I presume this is a writing error.
Line 342. RE broad host range in birds. It is likely broad, yet within certain clades of birds. Eg. Many species, but all birds are passerines. Please expand.
Line 343. What is Suid HV1 – this is not defined.
Line 375-377. This should be moved to be integratd into the discussion of oropharangeal route. It is misplaced here.
Reviewer 2 Report
This is an interesting study. Although this study is based on a small fragment of the herpesvirus genome, it provides some insight into the presence of herpesvirus in free living owls in temperate forests in Europe. In general, the study is well designed, and paper is well written.
- The authors used a consensus heprepsvirus PCR that amplifies a small fragment of the DNA polymerase gene potentially from all herpesviruses. However, the following factors that affect the sensitivity and efficiency of this PCR must be discussed and take into consideration when drawing conclusions.
- it is not clear whether this PCR is sensitive enough to detect all types of herpesvirus presents in Owl, mouse or other rodent samples. Deep sequencing of original samples may be necessary before arriving conclusions that “…. no rodent HVs were detected in owls…” in lines 37 and 360 .
- 3. lines 132-137: Please clarify how many swabs were pooled for each PCR, whether oropharyngeal swabs were pooled with cloacal swabs. In the discussion, clarify how pooling may have affected the sensitivity of the PCR.
- When testing cloacal swabs, inhibitory factors in fecal samples could affect the sensitivity of the PCR as well. Provide Clarification for this in the discussion.
- Line 135-137: state the percentage of the tissue homogenate prepared (weight/volume). Was it same for all tissue homogenates?
- Explain why Viral RNA Mini Kit was used instead of a dedicated DNA extraction kit? Is efficiency of DNA extraction same for RNA and DNA extraction kits? Did the use of RNA extraction kit affect the sensitivity of detection? Please explain.
Other comments.
- 137: State speed of centrifugation in ‘relative centrifugal force’ or g instead of ‘rpm’
- 1: PCR results of cloacal swabs are not stated in the results section, 3.1.
- Lines 297, 302, 391: Use ‘Partial sequence of DNA polymerase gene’ instead of “… HV DNA polymerase gene detected….” Since only a small fragment of the DNA polymerase gene was amplified by PCR and sequenced.
- Line 328: use ‘serotyping’ instead of serotypization
